# Modification of the MONERIS Nutrient Emission Model for a Lowland Country (Hungary) to Support River Basin Management Planning in the Danube River Basin

**Zsolt Jolánkai *** , **Máté Krisztián Kardos and Adrienne Clement**

Department of Sanitary and Environmental Engineering, Budapest University of Technology and Economics, 1111 Budapest, Hungary; kardos.mate@epito.bme.hu (M.K.K.); clement.adrienne@epito.bme.hu (A.C.)
* Correspondence: jolankai.zsolt@epito.bme.hu; Tel.: +36-1-463-2955

**Abstract:** The contamination of waters with nutrients, especially nitrogen and phosphorus originating from various diffuse and point sources, has become a worldwide issue in recent decades. Due to the complexity of the processes involved, watershed models are gaining an increasing role in their analysis. The goal set by the EU Water Framework Directive (to reach "good status" of all water bodies) requires spatially detailed information on the fate of contaminants. In this study, the watershed nutrient model MONERIS was applied to the Hungarian part of the Danube River Basin. The spatial resolution was 1078 water bodies (mean area of 86 km$^2$); two subsequent 4 year periods (2009–2012 and 2013–2016) were modeled. Various elements/parameters of the model were adjusted and tested against surface and subsurface water quality measurements conducted all over the country, namely (i) the water balance equations (surface and subsurface runoff), (ii) the nitrogen retention parameters of the subsurface pathways (excluding tile drainage), (iii) the shallow groundwater phosphorus concentrations, and (iv) the surface water retention parameters. The study revealed that (i) digital-filter-based separation of surface and subsurface runoff yielded different values of these components, but this change did not influence nutrient loads significantly; (ii) shallow groundwater phosphorus concentrations in the sandy soils of Hungary differ from those of the MONERIS default values; (iii) a significant change of the phosphorus in-stream retention parameters was needed to approach measured in-stream phosphorus load values. Local emissions and pathways were analyzed and compared with previous model results.

**Keywords:** MONERIS; nitrogen; phosphorus; diffuse nutrient emission; empirical modeling; river basin management plan of Hungary

## 1. Introduction

### 1.1. The Nutrient Problem

Anthropogenic activities such as intensive agriculture and communal and industrial wastewater discharges lead to nutrient over-enrichment in waters. Nitrogen and phosphorus play primary roles in the deterioration of the water quality of rivers and lakes. Many countries have implemented strict measures to protect the status of waters. Two examples are the Clean Water Act in the US and the Water Framework Directive in Europe. According to the latter, EU (European Union) member states are obliged to reach a "good" status of each water body by 2027 at the latest. River basin management plans (RBMPs) prepared in a six year cycle describe both the current status and the measures that must be taken to reach this goal.

In Hungary, the ecological status of surface waters is generally worse than the EU average. In the first RBMP, approximately 6% of the surface water bodies of Hungary were classified as poor, and 35% were classified as moderate based on physicochemical conditions [1,2]. In these water bodies, the inadequate status was predominantly due to the nitrogen and phosphorus concentrations, with a stronger influence of the latter.

Due to the various sources and pathways from/through which nitrogen and phosphorus can reach surface and subsurface waters, it is difficult to choose the optimal mitigation option. Watershed models provide a useful tool with which to tackle this problem [3].

There are several available watershed models with the capacity to model nutrient emissions to surface waters. The scale of models spreads from event-based dynamic plot scale models to static, annual time-scale, basin-scale models [4]. Model review studies have been carried out by multiple authors to compare and assess the accuracy of these models [5–7]. The most widespread models in this field include AGNPS, AnnAGNPS, ANSWERS, CASC2D, DWSM, HSPF, KINEROS, MIKE SHE, PRMS, SWAT, and MONERIS. The review of phosphorus emission models led the authors of the paper to the conclusion that it would be favorable to use models that use parsimonious approaches, where yearly time steps are used with a relatively low number of input parameters combined with a stochastic framework [7]. The EUROHARP project aimed specifically to give a better insight into the similarities and differences among watershed-scale nutrient emission models and stream water quality models (including NL-CAT, REALTA, and NOPOLU, among others) [8]. River retention estimates of the separate approaches were compared, and it was found that the variability of the estimates was large, and therefore it increased the uncertainty of the model predictions. A most recent study compared three frequently used nutrient emission models of the Danube basin for 18 ICPDR (International Commission for the Protection of the Danube River) regions [9]. The study aimed to compare and conclude the results of the models and not to assess the models themselves. The study revealed that all three models (SWAT, MONERIS, GREEN) were capable of estimating yearly nutrient loading; they showed coherent results with each other, but the GREEN model consistently overestimated TP (Total Phosphorus) loads.

The model system MONERIS was developed in the Leibnitz Institute of Freshwater Ecology and Inland Fisheries in Berlin [10]. It is a nutrient emission model developed to achieve three goals: (i) to identify the sources and pathways of nutrient emissions of AUs (spatial units for which input data are aggregated), (ii) to analyze nutrient transport and retention in river systems, and (iii) to create a framework to assess management alternatives for river systems. It has been used for German river catchments [11], for other catchments in Europe (Vistula, Po, Odra) [12] and outside Europe [13], and for the whole Danube catchment [14,15].

## 1.2. Aims of the Study

Even though a country-scale assessment of phosphorus emissions has already been carried out in Hungary [16], the same is not true for nitrogen. Nitrogen loads have not been estimated with pathway-specific modeling. Neither has an integrated point-/diffuse-source emission model been applied for either of the substances.

The MONERIS model was chosen for our study as it has been previously used and validated in the Danube Basin, including Hungary, and because it seemed to have a balance between model complexity and data demand, as was also concluded by other authors [17]. Even though the MONERIS model has been used in many river basins, it has to be stressed that this is a semi-empirical model, with tens of parameters set by data collected in Germany and wider Europe. They should be applied with caution in regions where the hydrological, hydrogeological, soil conditions, land use structures, and infrastructures characterizing of the calibration area are significantly different [17].

This study aimed to

- determine the degree to which the model system MONERIS is capable of estimating nutrient fluxes in a lowland country such as Hungary;

- identify the model components (equations, parameters) that had to be adjusted to better describe processes in the study area;
- give country-wide, waterbody scale estimates for nitrogen and phosphorus loads of various pathways.

## 2. Materials and Methods

### 2.1. Study Area

The analysis was carried out across the whole area of Hungary, which is located within the Carpathian Basin and has a total area of 93,000 km$^2$. The country is dominated by two flat, lowland areas with an average altitude below 200 m a.s.l. (the Alföld in the southeast, and the Kis-Alföld in the northeast). There are also several hilly and mountainous regions (altitudes still below 1000 m a.s.l.) in the central western, southern, and the northeastern parts of the country. The country lies entirely within the Danube Basin, and it can be divided into four subcatchments: the Danube, Tisza, and Drava River catchments, and Lake Balaton subcatchment (Figure 1). The Tisza River subcatchment covers the largest area of the country, while Lake Balaton is the most important standing water in the country. The average runoff rate is small (10–50 mm in the lowlands, and 50–200 mm in the hilly/mountainous parts [18], which is due to the relatively low precipitation, high evapotranspiration, and small reliefs, while the inflowing river flows are high (~1200 mm) [19]. Besides the natural river bodies, it is relevant to mention that the lowlands of Hungary are drained by an extensive network of artificial channels (total length is approx. 42,000 km) to protect the farmlands from excess water.

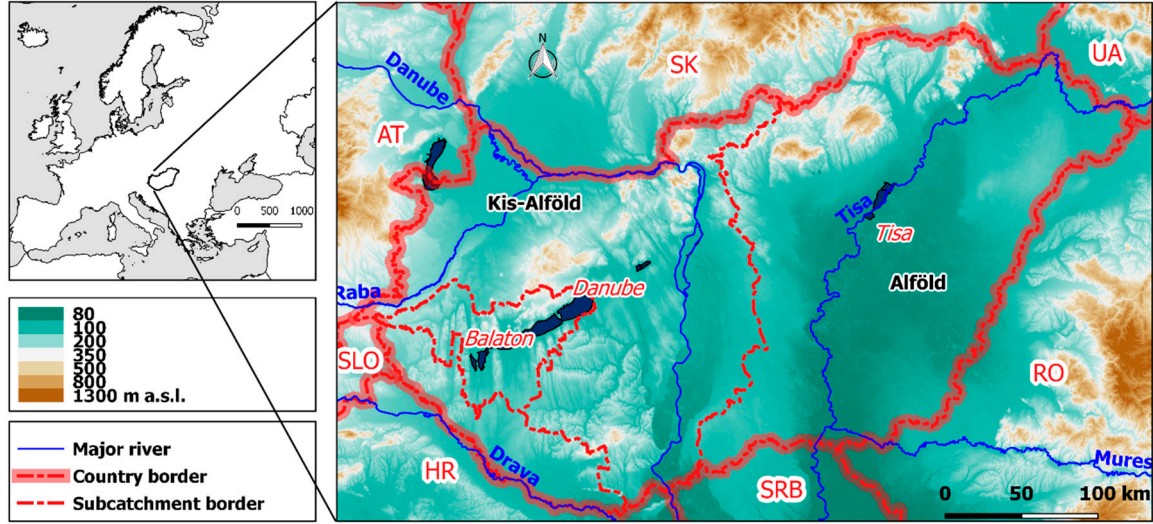

**Figure 1.** Overview of Hungary and its main rivers and subcatchments.

Hungary lies in the temperate climatic zone and its climate is very erratic, as three different climates have a strong effect on it: the oceanic climate; the dry continental climate, with extreme temperatures; and the Mediterranean climate, with dry summers and wet winters. Any of these can rule the climate of the Carpathian Basin for shorter or longer periods within the year [20]. The long-term average precipitation in Hungary is between 500 and 750 mm with a strong spatial variability, with the wettest region in the southwest and the driest in the central eastern part of the lowland regions. The most precipitation occurs in the mountainous regions throughout the country. Within this spatial distribution, the Drava and Danube catchments receive significantly more precipitation than the Tisza catchment in the eastern part of Hungary [20]. The wettest season is the summer (June being the wettest with around 70 mm) and the driest season is the winter (below 30 mm in February). The yearly average annual precipitation is tending towards a slight decrease, with a 10% decrease in the last

century. Extremes both in precipitation and temperature are frequent phenomena, often occurring within a single year in the same region (e.g., the year 2000).

In the upmost 10 m, the country is predominantly (ca. 85%) covered by loose sediments, mostly loess and sand, while clay also makes up a significant proportion. Solid rocks are only present in small mountainous areas, in the northeastern part of the country and near the Balaton, while porous limestone regions are slightly more frequent in the mountainous regions. The majority of the country (especially the lowlands) is covered by good, fertile soils suitable for agricultural production [21].

*2.2. Model Description*

MONERIS is an empirical, spatially semi-distributed, time-aggregated catchment model. The smallest spatial unit of calculation is an *administrative unit* (AU), which comprises one or more hydrologically interrelated river stretches and their direct catchment area. Since only a tiny part of the AU is monitored (either for water quantity or for quality), the term *catchment* is used throughout this paper for the catchments of monitoring points comprising one or more conterminous, hydrologically interrelated AUs.

In this section, only those parts of the model that were investigated or adjusted for the Hungarian conditions are described. For further information on the model's structure and equations, the reader is referred to Reference [22], which is also the source for the relationships described below.

### 2.2.1. Flow Components/Water Balance

The MONERIS model assumes seven main water pathways, according to Equation (1).

$$Q = Q_{atm} + Q_{surf} + Q_{sm} + Q_{tile} + Q_{gw} + Q_{urb} + Q_{ww} \tag{1}$$

where $Q$. is the total runoff in the AU (model input), $Q_{atm}$ is runoff from precipitation falling directly on open water surfaces, $Q_{sm}$ is runoff from snowmelt, and $Q_{gw}$ is the natural groundwater flow. $Q_{urb}$ is runoff (originating from precipitation + wastewater) from unsewered urban areas and $Q_{ww}$ is wastewater discharge (model input). $Q_{surf}$ is the surface runoff and $Q_{tile}$ is tile drainage flow. Flow is expressed in m$^3$/s.

$$Q_{surf} = k_{w1} \cdot Q^{k_{w2}} \tag{2}$$

$$Q_{tile} = A_{tile}(k_{w3} \cdot P_s + k_{w4} \cdot P_w) \tag{3}$$

where $A_{tile}$ is the area of tile-drained agricultural fields; $P_s$ and $P_w$ are summer and winter precipitation (mm) values, respectively; and $k_{w1}$, $k_{w2}$, $k_{w3}$, and $k_{w4}$ are model constants (-). $Q_{gw}$ is calculated as the difference between $Q$ and the six other terms.

$$Q_{gw} = Q - \left(Q_{atm} + Q_{surf} + Q_{sm} + Q_{tile} + Q_{urb} + Q_{ww}\right) \tag{4}$$

### 2.2.2. Subsurface Pathway of Nitrogen and Phosphorus

Concerning the subsurface fate of nitrogen, the MONERIS approach separates four hydrogeological types: unconsolidated porous media with deep or shallow groundwater, and consolidated rock with good or poor permeability. The equation for the calculation of groundwater nitrate levels is as follows.

$$C_{GW} = \left( \sum_{i=1}^{4} \frac{1}{1 + k_{n1i} \cdot R^{kn2i}} \cdot \frac{A_{HG,i}}{A_{AU}} \right) \cdot (C_{LWPOT})^{kn3} \tag{5}$$

where

$C_{GW}$ is the nitrate–nitrogen concentration of groundwater (mg N L$^{-1}$), $A_{HG,i}$ is the area of the catchment of hydrogeological class $i$ (km$^2$), $R$ is the long-term average recharge of the catchment (mm y$^{-1}$), $A_{AU}$ is the size of the AU (km$^2$), $C_{LWPOT}$ is the potential nitrate–nitrogen concentration of

the leachate (mg L$^{-1}$), $k_{n1i}$, and $k_{n2i}$ are model constants varying by hydrogeological type, and $k_{n3}$ is the model coefficient for denitrification.

Regarding subsurface phosphorus, the following five soil types are distinguished by the model: sandy soils, clayey soils, fen soils, bog soils, woodland, and open areas.

$$C_{GWAG}^{SRP} = \frac{\sum_{i=1}^{4} C_{GWi}^{SRP} \cdot A_i}{\sum_{i=1}^{4} A_i} \tag{6}$$

where $C_{GWAG}^{SRP}$ is the groundwater soluble reactive phosphorus (SRP) concentration for agricultural land (mg P L$^{-1}$); $C_{GWi}^{SRP}$ is the groundwater SRP concentration for sandy, loamy, fen, and bog soil types defined as model constants; and $A_i$ is the area of sandy, loamy, fen, and bog soils (km$^2$). In a second step, the average SRP concentrations in groundwater are calculated as an area-weighted average of agricultural and non-agricultural areas.

$$C_{GW}^{SRP} = \frac{C_{GWAG}^{SRP} \cdot A_{AG} + C_{GW\ WOOP}^{SRP} \cdot A_{WOOP}}{A_{AG} + A_{WOOP}} \tag{7}$$

where $C_{GW}^{SRP}$ and $A_{WOOP}$ are the groundwater SRP concentration and area of woodlands and open areas, respectively, and $C_{GW}^{SRP}$ is the groundwater SRP concentration (mg P L$^{-1}$).

### 2.2.3. Nutrient Retention in Tributaries

In the MONERIS model, in-stream concentration reducing processes such as settling, denitrification, etc. are considered to be aggregated under the term "retention". Retention coefficients are calculated as a function of the hydraulic loads, mean water temperature, and area-specific discharge (Equations (8)–(10)).

$$R_{TN} = \frac{1}{1 + k_{r1} \cdot e^{kr2 \cdot T} \cdot HL^{kr3}} \tag{8}$$

$$R_{DIN} = \frac{1}{1 + k_{r4} \cdot e^{kr5 \cdot T} \cdot HL^{kr6}} \tag{9}$$

$$R_{TP} = \frac{1}{1 + k_{r7} \cdot q^{kr8}} + \frac{1}{1 + k_{r9} \cdot HL^{kr10}} \tag{10}$$

where $R_{TN}$ is the retention coefficient for TN, $k_{r1}$ to $k_{r3}$ are the retention parameters for TN, $k_{r4}$ to $k_{r6}$ are the retention parameters for DIN (dissolved inorganic nitrogen), $HL$ is the hydraulic load for the water body (river flow/water surface, m y$^{-1}$), $T$ is the yearly average water temperature (°C), $R_{TP}$ is the retention coefficient for TP, $k_{r7}$ to $k_{r10}$ are the retention parameters for TP, and $q$ is the area-specific runoff (l s$^{-1}$ km$^{-2}$).

Validation of the model is generally achieved using surface water quality monitoring points, where sufficient data are available both for flow and water quality parameters. In the current case, total nitrogen and total phosphorus measurements were the critical parameters for validation of the nutrient emission estimation, as their measurements are generally scarce both spatially and temporally.

### 2.2.4. Modeling Environment

The calculation of local emissions according to the equations described in the model version 2.14.1 [22] was carried out in a Microsoft Excel environment [17]. For calibration purposes, the cumulative river load calculations along the river hierarchy and the river retention were calculated in the MATLAB environment. MATLAB and R [23] were used to process water quality data. The ArcMap 10.1 model version was used for the preparation of spatial data. This software, along with ArcMap and QGIS, was used to visualize the results.

*2.3. Preparation of Model Input Data*

In the present study, the MONERIS model was applied to the Hungarian part of the Danube River Basin for two subsequent 4 year periods, 2009–2012 and 2013–2016. However, the input data—depending on availability—comprised shorter or longer periods. Only some of the input data were updated between the two periods.

2.3.1. Delineation of AUs and Calculation of Their Basic Characteristics

Extensive use of spatial data was necessary to initialize the model. Model AUs were water bodies determined by the RBMP of Hungary. This meant a total of 1078 catchments, 189 of which belonged to lake water bodies, and the rest of which belonged to river water bodies. The average size of the catchments was 83 km$^2$ (median was 51.7 km$^2$) and the maximum size was 1166 km$^2$ (Lake Balaton), the smallest being only 0.3 km$^2$.

Concerning hydrometeorology, the National Meteorological Service provided long-term mean precipitation values for the years 1981 to 2010 in a grid-based format. Monthly actual evapotranspiration for the period 2000 to 2009 was also involved [24,25]. Daily precipitation and mean temperature data for 245 stations were provided by the General Directorate of Water Management, Hungary, for the period 1991–2016. These data were interpolated using inverse distance-weighted interpolation and kriging.

Topographical properties such as elevation and slope were calculated by processing digital elevation data [26]. Water network characteristics were calculated based on the River Water Body and River Water Segment spatial databases [27]. Land use data were included based on the Corine Land Cover Database [28]. Soil hydrogeological and topsoil physical properties were processed using the AGROTOPO [29] and DOSOREMI [30] databases.

2.3.2. Runoff

As Hungary lacks a country-wide hydrological model, runoff values were estimated using the following five-tiered approach. For gauged headwater AUs, time series' mean values were used. For ungauged AUs of gauged catchments, long-term mean flow at the catchment outflow point was spatially weighted by the difference between long-term mean precipitation (P) and evapotranspiration (ET). On some AUs (especially the ones along the Tisza River), long-term values of evapotranspiration exceeded those of precipitation; for such catchments, long-term mean flow was spatially weighted by long-term mean runoff values [31]. On ungauged catchments, where both sandy and limestone areas occupied less than 50% of the catchment area, the runoff was estimated by establishing a linear regression between model period mean flow of gauged catchments and their channel length values. For AUs where neither of the above methods were applicable, long-term mean runoff values [31] were corrected by the proportion of the investigated periods and the long-term rainfall.

2.3.3. Soil Loss and Nutrient Surplus

The universal soil loss equation [32] was used for the mean annual soil loss estimation in each of the AUs. C and R factors were determined based on the European scale JRC maps [33] and K factor maps were prepared by the Research Institute of Soil Sciences and Agricultural Chemistry [34]. L and S factors were determined from a 50 m resolution hydrologically corrected digital elevation model [35].

Yearly nutrient balance estimations were calculated for the period 1961–2016, according to the OECD (Organisation for Economic Co-Operation and Development) method used to calculate gross nutrient balance based on county-scale agro-statistical data on fertilizer inputs, harvested yields, and animal husbandry [36,37]. Soil nutrient conditions for each county were allocated to the catchments directly (Figure 2).

For a detailed description of the data sources and their processing, see Tables A1 and A2 in the Appendix A.

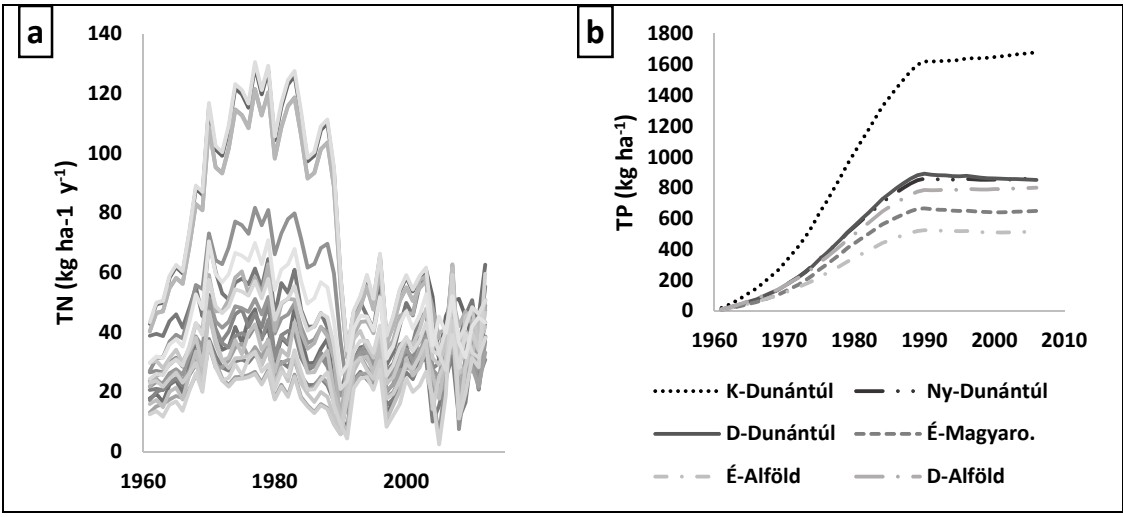

**Figure 2.** Total nitrogen (TN) surplus by the Hungarian counties (each county is represented by one line) (kg ha$^{-1}$ y$^{-1}$) (**a**) and accumulated total phosphorus (TP) by the Hungarian regions (kg ha$^{-1}$ É,D,K,NY stand for north, south, east and west) (**b**).

### 2.3.4. Nutrient Emission Pathways

Concerning atmospheric deposition, the EMEP (European Monitoring and Evaluation Programme) model results were used for oxidized and reduced nitrogen compounds [38]. Values increased from southeast to northwest from 360 to 570 mg N/m$^2$ for NO$_x$, and from 460 to 850 mg/m$^2$ from east to west for reduced nitrogen forms. Values for the 2013–2016 period were around 3% higher, presumably due to the higher precipitation values.

Concerning point-source loads, the national wastewater information system [39] data were processed to ascertain wastewater discharge and emission loads. Each operating wastewater treatment plant's (WWTP) emissions were linked to the water body it discharges into. Discharge, TN, and TP load values were used. There was a reduction in country-wide emission values between the two modeled periods: 9000 t y$^{-1}$ to 7384 t y$^{-1}$ and 1028 t y$^{-1}$ to 852 t y$^{-1}$ for total nitrogen and total phosphorus loads, respectively.

Urban diffuse emissions estimations require extensive information about the separate and combined sewer network, the connected population, etc. Detailed data were not available for the whole country; only county-scale rough estimates were available from the National Water Directorate and the Hungarian Central Statistical Office (KSH). Information about inhabitants connected to sewer systems and WWTPs was available from the national sewage information system [39].

### 2.4. Water Quality Data Used for Calibration and Validation Purposes

#### 2.4.1. Groundwater Well Data

To support the recalibration of nitrogen retention parameters (Section 2.2.2) and to review the average phosphorus levels in groundwater, a monitoring database of the shallow groundwater was processed. Water quality data were gathered for the period 2004–2018 for the whole country. Wells within 500 m of artificial areas and within 1500 m of larger rivers were excluded from the analysis (Figure 3). The rest of the wells were overlapped with the Corine Land Cover Map [28], the map of topsoil USDA classes [30], the hydrogeological type [29], and the groundwater depth map [40]. Using the spatial information of the maps, wells were classified into four nitrogen and four phosphorus categories (Tables 1 and 2).

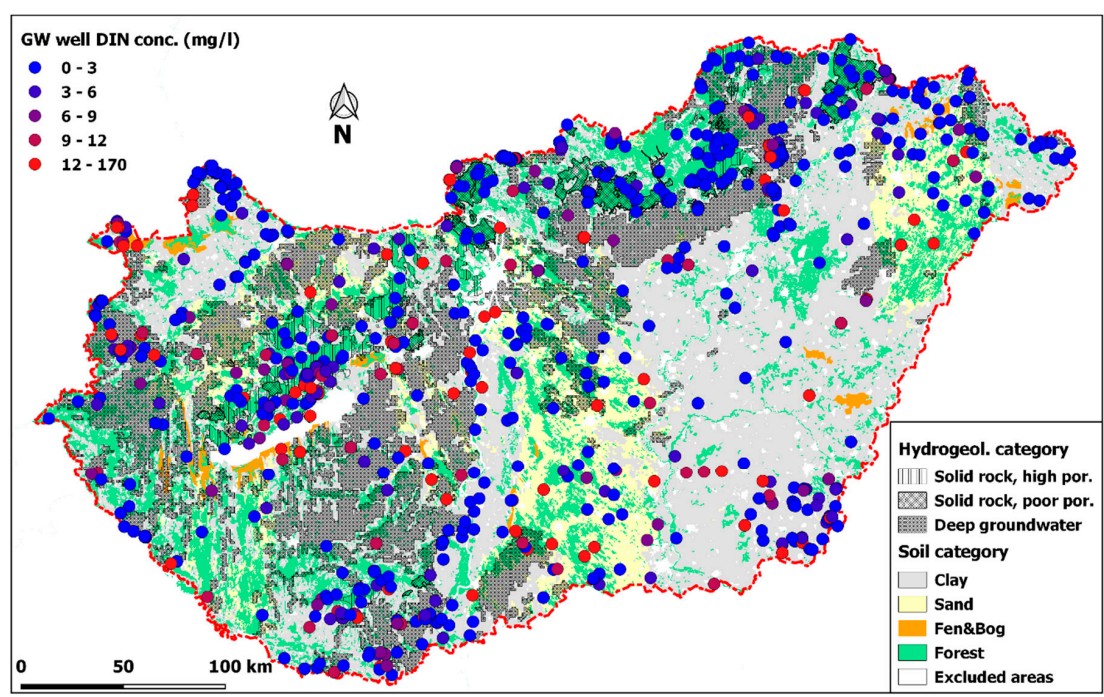

**Figure 3.** Shallow groundwater wells used for model calibration and geological and soil types according to MONERIS classification (GW – Groundwater, DIN – dissolved inorganic nitrogen).

**Table 1.** Well categories for nitrogen.

| Well Category for Nitrogen | Groundwater Depth | Soil Geology | Number of Wells |
|---|---|---|---|
| Unconsolidated rock, shallow groundwater | <3 m * | any other | 1022 |
| Unconsolidated rock, deep groundwater | >3 m * | any other | 55 |
| Solid rock, high porosity | any | limestone | 69 |
| Solid rock, impermeable | any | granite, andesite | 36 |

* the horizon for the division between the two groundwater depth classes was not defined in the literature. For this calculation, the threshold was set as 3 m.

**Table 2.** Well categories for phosphorus. SL = sandy loam; LS = loamy sand.

| Well Category for Phosphorus | Land Cover | Soil Texture | Soil Physical Category | Number of Wells |
|---|---|---|---|---|
| Sandy soils | agricultural | SL, LS, or sand | not peat or mull | 81 |
| Clayey soils | agricultural | any other | not peat or mull | 482 |
| Fen & bog soils | any | any | peat or mull | 5 |
| Woodland & open land | forest & seminatural | any | not peat or mull | 250 |

Time series mean DIN and PO4-P concentrations were calculated for each well. Category mean and median values were plotted on the histogram, and category median values were used for further processing.

### 2.4.2. Surface Water Quality

The monitoring concept in Hungary, but also in other countries in Europe, uses different ways to monitor general water quality status (averages), to identify concentration maxima at point-source discharges or to collect information about the state of smaller river sections. For the latter, monthly measurements are organized for a sequenced (1 year) period, and then repeated in the next period of revision of the water quality of the rivers. Besides this, there are other shifts of the monitoring points according to the practice in Hungary. The frequency of monitoring changes along time at other points, which means a change in the reliability of load estimation.

For calibration and validation purposes, hydrological data were obtained from the National Hydrological Database (MAHAB). Water quality data from the surface water module of the National Environmental Information System (OKIR–FEVISS) were also processed and utilized. Monitoring points for the current analysis is shown on Figure 4.

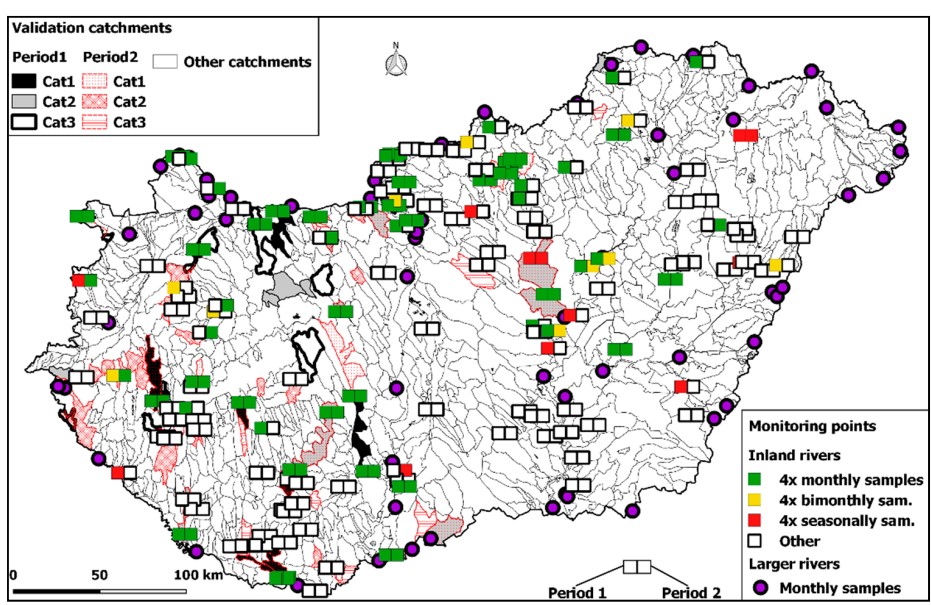

**Figure 4.** Surface water flow and quality monitoring stations.

### 2.5. Adjustments of the Model Parameters and Structure

#### 2.5.1. Runoff Separation with Digital Filter

As surface runoff and groundwater flow are characterized by different concentrations of nitrogen and phosphorus, it is important to estimate their proportions as accurately as possible. In the MONERIS model, the groundwater flow was estimated using Equation (4). This equation is the result of subtracting all other flow components from the total AU runoff (model input), many of which are calculated with empirical formulas (see Equations (2)–(3)). Due to errors in the input data and the estimation formulas, the estimated groundwater flow can be unrealistically small.

In this study, groundwater flow proportion values calculated by the MONERIS equations were compared to those calculated by the digital-filter-based method [41] developed by Arnold [42]. For this purpose, catchments with a flow gauge (daily discharge data) described by monthly TN/TP measurements and consisting of fewer than three AUs were selected. For the sake of comparison, atmospheric runoff and urban runoff were considered to be part of the surface runoff component, while tile drainage and point-source discharge were considered to be part of the baseflow. The aim of this comparison was to get a view of the model's sensitivity of the nutrient loads to the division of surface and subsurface waters.

#### 2.5.2. Groundwater Nitrogen Concentration

Parameter calibration was carried out against median DIN (Dissolved Inorganic Nitrogen) concentrations (mg N L$^{-1}$) measured in the wells. As the accuracy and especially the spatial resolution of nitrogen surplus of agricultural sites are very poor, calibration simply aimed for the improvement of catchment groundwater nitrate levels against groundwater well averages. In the calibration process, the AUs were used as the domain for input data aggregation. This means that the calculation of groundwater recharge, surplus, and leachate values are carried out for the AUs. The AUs with groundwater well data were selected and included in the calibration dataset. Over 300 AUs were included and they covered all of the geological categories, but most AUs had varying geology in their

area; therefore all of the geological categories were calibrated simultaneously. The parameter fitting was carried out with the generalized reduced gradient (GRG2) non-linear optimization method [43]. The objective function for parameter fitting was the sum of square errors.

### 2.5.3. River Retention

Model improvement was carried out through the implementation of parameter fitting of the river nutrient retention parameters ($kr_1$–$kr_{10}$, Equations (8)–(10)) for all nutrient components described in the model. Optimization of the parameters was done with the Matlab software "fmincon" function, using the interior point algorithm [44]. For DIN and TP, the objective function was the sum of mean square error, while for TN the sum of relative errors was used. Calibration points were classified into three categories and all categories were taken into account with equal weights.

### 2.6. Model Validation

As Hungary is a "downstream country", many of the available monitoring stations are placed on large rivers with transboundary catchments (the Danube, the Tisza, and the Drava, to name just the biggest ones) that are of no use if the aim is to calibrate or validate processes happening inside the country. Only monitoring stations for which the entire catchment falls within the country borders can be considered for validation. The available monitoring data were divided into three classes based on the reliability of yearly average loads calculated from the time series.

The most reliable class contained the data from stations with monthly sampling regimes (at least 40 measurements throughout each of the 4 year periods). The second group was formed from the data with at least bi-monthly sampling frequencies, whereas the third group contained the data from stations with seasonal measurements. Considering the fact that the number of stations represented only 5.75% of the total (modeled) catchments, the number of stations with adequate data was very limited (Table 3).

**Table 3.** The number of monitoring stations per uncertainty category and model period. Per. 1: 2009–2012; Per. 2: 2013–2016.

| Category | Yearly Sample Number | Per. 1 | Per. 2 | Total (w/o Overlap) |
|----------|----------------------|--------|--------|---------------------|
| Cat. 1 | 10+ | 8 | 16 | 20 |
| Cat. 2 | 6–9 | 16 | 18 | 32 |
| Cat. 3 | 4–5 | 9 | 9 | 18 |
| Cat. 1–3 | | 33 | 43 | 62 |

Weekly sampling is only carried out at one of the studied stations, at the lower section of the Zala River, which is the primary tributary to Kis-Balaton wetland and Lake Balaton. For this reason, the Zala River catchment was the most important validation catchment in this study, while other mid-sized catchments with lower sampling size followed, such as Kapos, Zagyva, Babócsai-Rinya, Fekete-Víz, Marcal, and Lónyai-csatorna catchments. These catchments also collect daily flow monitoring data that were helpful to increase the accuracy of the yearly average estimations.

In model validation, the annual average loads (t/y) were compared to the loads calculated by the model. The scale of the catchments and therefore the loads also varied with several orders of magnitude. This gave a good indication of the model's validity.

## 3. Results

### 3.1. Flow Components/Water Balance

Most of the AUs fell in measured catchments and thus were assigned to Tier 1 (43 AUs) or Tier 2 (537 AUs). In Tier 3, 208 AUs were identified. The regression equation (Tier 4, 276 AUs) was mainly

used along the larger rivers: Danube, Drava, Upper Tisza, Rába, and Nádor Channel (Figure 5a). The relationship (Equation (11)), Figure 5b) proved to be indicative ($R^2 = 0.93$, p < 0.0001).

$$Q = 0.003 \cdot L - 0.030 \tag{11}$$

where $Q$ is the flow (m$^3$ s$^{-1}$) and $L$ the channel length (km). Tier 5 was used only with a 14 AUs. The average difference between the actual period and long term mean runoff was −2.8 and 27.6 mm$^{-s}$ for Periods 1 and 2, respectively.

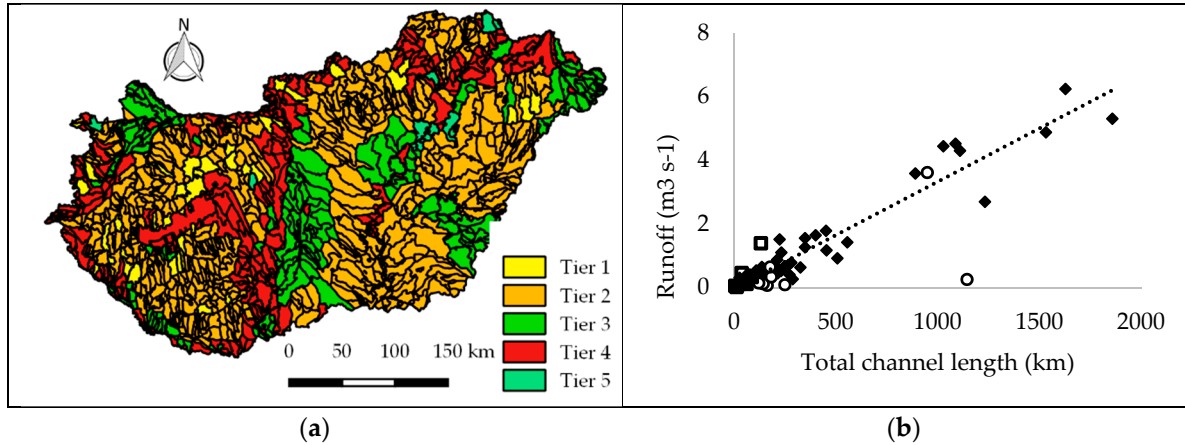

| (a) | (b) |

**Figure 5.** (**a**) Methods used for runoff estimation throughout the country. (**b**) measured average annual runoff vs. total channel length on the catchment. Hollow rounds: outliers from sandy catchments; squares: karstic areas.

The groundwater flow proportions calculated using different approaches showed strong similarities (Table 4). According to the MONERIS method, mean groundwater flow (total subsurface flow) share values were 0.73, 0.72, and 0.78 for the slope categories 0%–1%, 1%–5%, and >5%, respectively. According to the digital filter method, mean baseflow index values were 0.79, 0.68, and 0.64 for the same slope categories, respectively. Because surface runoff is higher on areas with higher slope (e.g., see Reference [45]), the digital-filter-based groundwater flow separation provided a more realistic result (where measured data were available), and it was selected for use in the model calculations.

**Table 4.** Comparison of surface runoff and baseflow index values calculated for gauged catchments as determined by the MONERIS (MON) and the digital filter (DF) methods.

|  | Catchment | | Total | Surface Runoff | | GW Flow Share | |
|---|---|---|---|---|---|---|---|
| **Catchment Name** | **Area** | **Slope** | **Runoff** | **MON** | **DF** | **MON** | **DF** |
|  | **(km$^2$)** | **(%)** | **(m$^3$ s$^{-1}$)** | **(m$^3$ s$^{-1}$)** | **(m$^3$ s$^{-1}$)** | **(-)** | **(-)** |
| Arany Creek | 36 | 1.9 | 0.33 | 0.022 | 0.043 | 0.85 | 0.70 |
| Kenyérmezei Creek | 125 | 11.3 | 0.15 | 0.000 | 0.000 | 0.79 | 0.76 |
| Kígyós Channel | 594 | 0.8 | 0.25 | 0.010 | 0.002 | 0.73 | 0.78 |
| Tapolca Creek | 51 | 4.1 | 0.28 | 0.061 | 0.000 | 0.84 | 0.85 |
| Tetves Creek | 88 | 8.2 | 0.18 | 0.022 | 0.034 | 0.84 | 0.69 |
| Torna Creek | 176 | 7.2 | 0.56 | 0.099 | 0.035 | 0.84 | 0.82 |
| Únyi Creek | 172 | 9.2 | 0.29 | 0.019 | 0.077 | 0.73 | 0.62 |
| Villány–Pogányi c. | 202 | 5.6 | 0.26 | 0.054 | 0.077 | 0.80 | 0.62 |
| Zagyva Creek (upper) | 168 | 14.1 | 0.94 | 0.112 | 0.439 | 0.84 | 0.48 |
| Average | 179 | 6.9 | 0.36 | 0.04 | 0.08 | 0.81 | 0.70 |

*3.2. Groundwater Pathway of Nutrients*

Groundwater nitrogen and phosphorus concentrations showed a large variation between the wells. The distribution of the concentrations showed that the vast majority of the wells were within a narrow concentration range of relatively low concentration values (except soils with a small number of wells, Figure 6). It was assumed that higher concentrations represented local extremities due to contamination from unregistered sources. Due to their small number, their impact on surface water loads might be negligible. For this reason, median values (as opposed to the mean values) were used in application of the model.

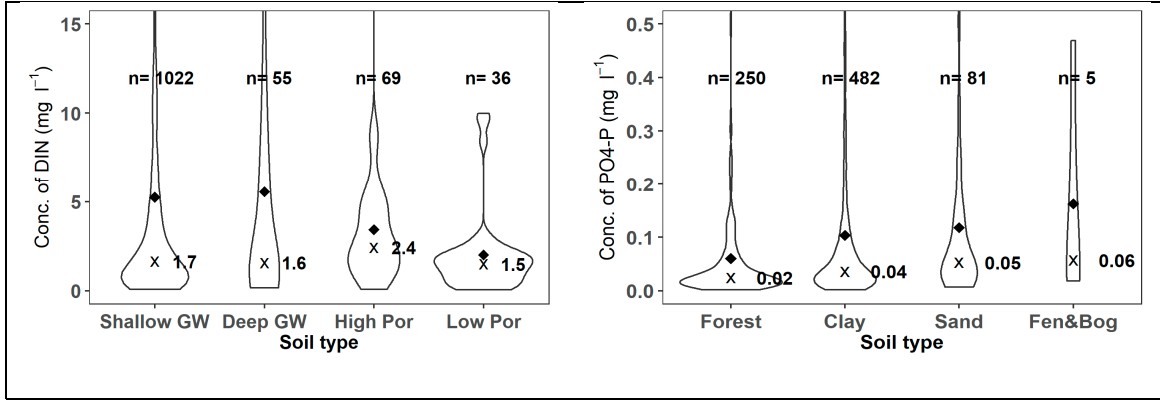

**Figure 6.** Kernel density plot showing the distribution of the measurements of dissolved inorganic nitrogen (**left**) and orthophosphate (**right**) concentrations measured in groundwater wells. Diamond: mean value, x: median value.

The calibration of groundwater nitrogen retention parameters resulted in new values (Table 5) that balanced out subsurface pathways (Figure 7) in Hungarian regions from hilly to mountainous parts of the country. The calibration resulted in a more robust estimation throughout the load scales in the river monitoring points, especially in the higher load range (Figure 7, Table 6). In the lower load range, slight overestimation was observed, while at the higher load values, some underestimation was still present.

Values for forests and open spaces were as advised by the original model dataset based on German monitoring results. For sandy agricultural soils, the values found in Hungarian wells were lower compared to the original model data (0.05 versus the original value of 0.1 mg L$^{-1}$). For clayey agricultural soils, the data were also identical to the original data. Several studies have stated that bog soils have higher nutrient values due to their high organic matter content [10]. The low number of wells situated in fen/bog soils of Hungary did support this statement. Due to a lack of representativeness, original P concentrations were not changed.

The adjustment of the groundwater nitrogen retention parameters improved the model's accuracy in terms of the coefficient of determination and in terms of the relative errors (Figure 7a,b, Table 6).

**Table 5.** Model constants before and after adjustment. The letters in the brackets correspond to those of chapter 2.1. Original: according to [21]; Adjusted: as considered in the present study. UC = unconsolidated; GW = groundwater.

| Process/Constant Name | Soil Category | Units | Original | Adjusted |
|---|---|---|---|---|
| **Subsurface Nitrogen** | | | | |
| Nitrogen constant 1 ($k_{n1}$) | UC rock, shallow GW | - | 2752 | 84.24 |
| | UC rock, deep GW | - | 68,560 | 7917 |
| | Solid rock, high porosity | - | 60.23 | 67.33 |
| | Solid rock, poor porosity | - | 78.54 | 99,787 |
| Nitrogen constant 2 ($k_{n2}$) | UC rock, shallow GW | - | −1.540 | −1.216 |
| | UC rock, deep GW | - | −1.959 | −3.750 |
| | Solid rock, high porosity | - | −0.903 | −1.124 |
| | Solid rock, poor porosity | - | −0.662 | −2.747 |
| Denitrification in topsoil ($k_{n3}$) | any | - | 0.6368 | 0.4340 |
| **Subsurface Phosphorus** | | | | |
| P conc. in groundwater $C_{GW}^{SRP}$ | Sandy agricultural soils | mg L$^{-1}$ | 0.10 | 0.05 |
| | Clayey agricultural soils | mg L$^{-1}$ | 0.03 | 0.03 |
| | Fen agricultural soils | mg L$^{-1}$ | 0.10 | 0.10 |
| | Bog agricultural soils | mg L$^{-1}$ | 0.50 | 0.50 |
| | Woodland and open areas | mg L$^{-1}$ | 0.02 | 0.02 |
| **River Retention** | | | | |
| TN constant 1 ($k_{r1}$) | | - | 4.74 | 78.8 |
| TN constant 2 ($k_{r2}$) | | - | 0.067 | −0.31 |
| TN constant 3 ($k_{r3}$) | | - | −1 | −0.53 |
| DIN constant 1 ($k_{r4}$) | | - | 8.58 | 14.39 |
| DIN constant 2 ($k_{r5}$) | | - | 0.067 | −0.06 |
| DIN constant 3 ($k_{r6}$) | | - | −1 | −0.51 |
| TP constant 1 ($k_{r7}$) | | - | 5.07 | 200 |
| TP constant 2 ($k_{r8}$) | | - | −1 | −9.69 |
| TP constant 3 ($k_{r9}$) | | - | 25.74 | 4.87 |
| TP constant 4 ($k_{r10}$) | | - | −1 | −0.89 |

**Table 6.** Calibration results for groundwater nitrogen retention parameters.

| | $R^2$ | | Absolute Relative Error | |
|---|---|---|---|---|
| | Original | Adjusted | Original | Adjusted |
| Cat. 1 | 0.78 | 0.95 | 52% | 21% |
| Cat. 2 | 0.74 | 0.92 | 23% | 31% |
| Cat. 3 | 0.86 | 0.98 | 82% | 17% |
| Cat. 1–3 | 0.48 | 0.95 | 46% | 25% |

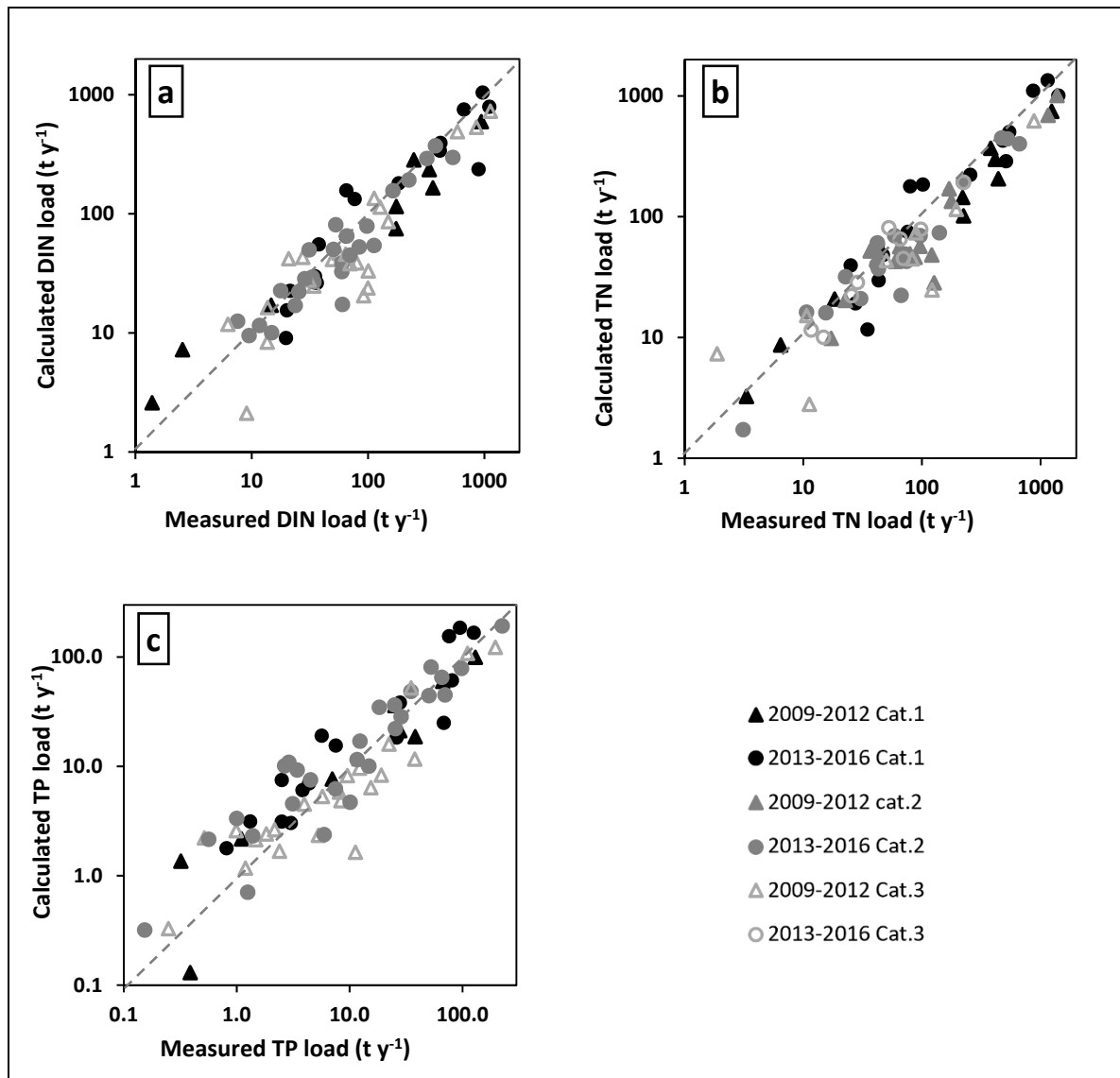

**Figure 7.** Measured and calculated loads at all monitoring stations. (**a**,**b**,**c***)* shows the validation results of DIN/TN and TP loads respectively.

### 3.3. River Nutrient Retention

The calibration of TN retention parameters resulted in a 1%–3% improvement of the sum of relative errors (Figure 7). Since these were rather small numbers, no strict recommendation can be made to use the adjusted retention parameters. In the case of DIN, the change in the retention was larger, but there was also very little improvement in the model accuracy ($R^2$ improved from 0.81 to 0.82 for the whole dataset). For TP, calibration of the retention parameters resulted in a significant improvement of model efficiency (Figure 7). Root mean square error changed from 1.12 to 0.67, from 0.88 to 0.48, and from 0.42 to 0.57 for load classes <5, 5–20, and >20 t y$^{-1}$, respectively. As a mean of all load categories, root mean square error changed from 0.81 to 0.57.

### 3.4. Model Validation

The validation of the model showed a relatively good agreement for all compounds (Figure 7, Table 7). The Category 3 validation dataset showed a larger relative error for all compounds except in the second period. For TN and DIN, the highest load range showed a slight underestimation. For TP,

the lower load range showed a stronger overestimation of loads, and the overall goodness of the estimation was slightly worse than for TN and DIN.

**Table 7.** Validation results of all compounds for the two periods.

| | Per. 1 | | | Per. 2 | | | Per. 1 + Per. 2 | | |
|---|---|---|---|---|---|---|---|---|---|
| Category | DIN | TN | TP | DIN | TN | TP | DIN | TN | TP |
| **Square of Pearson correlation coefficient ($R^2$)** | | | | | | | | | |
| Cat. 1 | 0.93 | 0.94 | 0.95 | 0.77 | 0.89 | 0.78 | 0.79 | 0.85 | 0.74 |
| Cat. 2 | 0.99 | 0.98 | 0.93 | 0.89 | 0.94 | 0.85 | 0.96 | 0.97 | 0.89 |
| Cat. 3 | 0.98 | 0.99 | 0.85 | 0.94 | 0.94 | 0.66 | 0.97 | 0.98 | 0.72 |
| *Cat. 1–3* | *0.96* | *0.97* | *0.91* | *0.86* | *0.92* | *0.76* | *0.9* | *0.93* | *0.78* |
| **Mean relative error** | | | | | | | | | |
| Cat. 1 | 0.76 | 0.35 | 1.18 | 0.49 | 0.48 | 1.04 | 0.6 | 0.44 | 1.09 |
| Cat. 2 | 0.48 | 0.39 | 1.01 | 0.41 | 0.35 | 1.41 | 0.44 | 0.37 | 1.25 |
| Cat. 3 | 3.13 | 1.07 | 4.36 | 0.37 | 0.26 | 1.14 | 2.23 | 0.78 | 3.19 |
| *Cat. 1–3* | *1.45* | *0.6* | *2.18* | *0.43* | *0.36* | *1.2* | *1.09* | *0.53* | *1.84* |

## 4. Discussion

### 4.1. Comparison of Nutrient Load Estimation Results on Gauged Catchments with Different Baseflow Separation Methods, Using the MONERIS Model

Baseflow indices (BFI, the ratio of the average baseflow to average total runoff) were close to each other in seven of the sub-catchments (<5% diff.), while in seven cases, the MONERIS method overestimated BFI by over 10%, and in two catchments it underestimated BFI by over 10%. The total average BFI was calculated to be 0.78 by MONERIS, while it was estimated to be 0.71 by the digital filter method.

Even though the baseflow indices calculated by the two methods differed significantly, the loads in the river at the monitoring points differed by less than 4% for all of the substances. For nitrogen, this can be explained by the fact that concentrations in the surface and subsurface pathways did not differ significantly. In the case of total phosphorus, this can be explained by the small share of these two pathways combined in total diffuse emissions (less than 15% combined).

Differences were more substantial with regard to the distribution of the emissions via the surface and subsurface pathways (Table 8, Figure 7).

**Table 8.** Total nitrogen and total phosphorus loads in unique pathways before and after the correction of the groundwater flow values.

| | Surface Runoff | | Groundwater | | Total Diffuse |
|---|---|---|---|---|---|
| | (t y$^{-1}$) | Ratio to Total (%) | (t y$^{-1}$) | Ratio to Total (%) | (t y$^{-1}$) |
| **Total nitrogen** | | | | | |
| Original | 16 | *3.4* | 242 | *52* | 468 |
| Adjusted | 52 | *12* | 185 | *41* | 447 |
| Difference (%) | *69* | | *31* | | *4.7* |
| **Total phosphorus** | | | | | |
| Original | 1.6 | *2.6* | 5.7 | *9.0* | 63 |
| Adjusted | 5.4 | *8.3* | 4.1 | *6.3* | 65 |
| Difference (%) | *70* | | *36* | | *3.5* |

For nitrogen, the share of surface runoff in total diffuse nitrogen emissions increased from 3.4% to 11.7%, while groundwater share dropped around 10%. This finding raised some concern regarding the

surface runoff pathway as it leveled up with urban diffuse emissions (56.2 and 46.9 t y$^{-1}$ respectively), and combined with agricultural erosion, it almost leveled up with groundwater.

For total phosphorus, the share of surface runoff increased from 2.6% to 8.3%, while the share of groundwater decreased from 9% to 6.3%. This was not a dramatic change, especially given that agricultural erosion dominates the pathways followed by urban runoff emission. According to these results, however, dissolved phosphorus in agricultural runoff was three times more important than the original model results indicated. Agricultural runoff would have been even more significant if agricultural soils were closer to phosphorus saturation, because dissolved P levels in surface runoff increase nonlinearly with P saturation of soils [15,46].

It is evident therefore that the right baseflow index or the right portion of surface and subsurface runoff is important for the determination of the share of surface and subsurface emission pathways. A more accurate estimate for this as made here by using digital filters directly where possible, e.g., where measured runoff was available at the outlet of an analytical unit. This would improve the accuracy of the load estimates and possibly highlight the need for measures to be taken to control surface runoff and erosion on agricultural areas.

### 4.2. Subsurface Processes

According to the results, nitrogen retention in subsurface pathway is higher in high porosity consolidated rocks and lower in unconsolidated rocks with shallow groundwater than in previous studies (Figure 8). The latter can be caused due to the very low runoff from lowland parts of the country. In the Alföld area (Figure 1), there is a concentration increase due to two separate phenomena: (1) the concentration increase due to the re-evaporation of groundwater and (2) the higher background concentrations in the groundwater due to geochemical reasons.

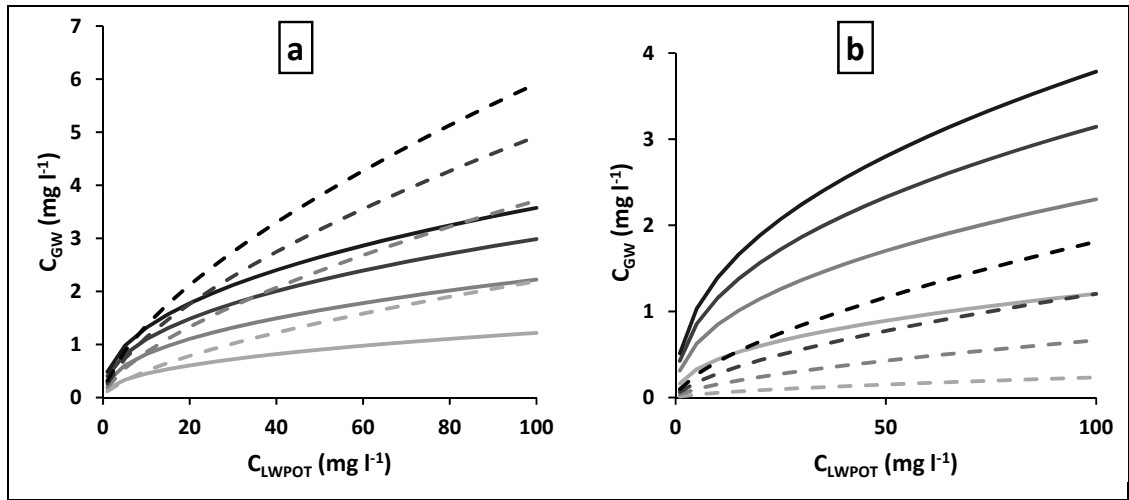

**Figure 8.** Groundwater nitrogen concentration vs. potential leachate concentration and their dependency of annual recharge using old and new retention parameters. (**a**) Consolidated rocks with high porosity; (**b**) unconsolidated rock with shallow groundwater. Dashed line: original parameters; continuous: adjusted parameters. Recharge values 10, 20, 30, 40 mm y$^{-1}$ from lighter to darker colors.

### 4.3. River Retention

Total phosphorus load values were systematically underestimated at all scales with the original parameters. Possible causes might have been the underestimation of the emissions (possibly higher loads from the agricultural areas) or disregarding of the internal loads from river sediments. The latter is well-known in shallow lakes [47] and can be observed after external load reduction [48]. Generally, phosphorus retention is regulated by a variety of physical, chemical, and biological factors, by the physical processes in streams, such as flow velocity, discharge, and water depth, are dominated. Abiotic

sorption reactions controlling P retention in streams are the same as in wetlands; however, long-term storage of P in stream sediments is inhibited by the rapid mobilization and transport that occurs during storm events [49]. TP retention is more sensitive to the specific total runoff than to the hydraulic loads (Figure 9).

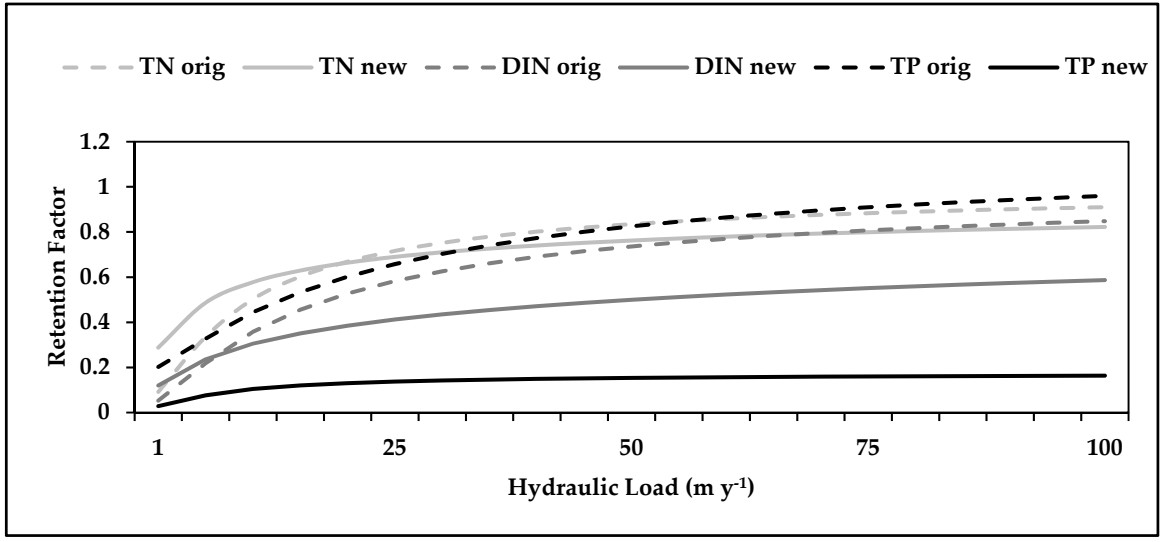

**Figure 9.** Effect of original and adjusted retention parameters on the retention factor as a function of the hydraulic load (specific total runoff for TP retention was $1\,\mathrm{l\,s^{-1}\,km^{-2}}$).

It was a strong result, however, that the change of tributary retention of the catchments resulted in a much better fit at almost every monitoring point. However, we noted that this was not true for the lower load ranges, where decreased retention resulted in an overestimation of the loads. The other possible sources of TP that the model missed with the original retention values might have been higher diffuse-source loads and point-source loads, but these would differ among the study catchments, and it is therefore unlikely that this was the real reason. Another possible cause for the systematic underestimation of river TP loads may have been the underestimation of the sediment delivery ratio, as sediment transports the largest fraction of TP to the river outlets.

This latter, however, is in conflict with previous findings on this topic, and the overestimation of the total phosphorus loads in the lower load ranges suggested that the retention equation adjustment might compensate other loads not taken into account in the model. From the current analysis, it was difficult to make a conclusion on this issue; it should be subjected to further investigation on possible internal loads from river sediments or other pathways, including sediment delivery. The former was anticipated as there was a significant drop in point-source (PS) loads due to the increase in the number of wastewater treatment plants in the country, and the efficiency of sewage treatment. The drop in PS emissions is generally followed by the release of phosphorus from the sediments, as was found for lakes and wetlands [50], but this might also be the case for river sediments.

## 4.4. Model Validation in Context of the Literature

It is well known that the number of measurement data needed to produce reliable yearly averages differ for different compounds and different sampling locations [51]. Total nitrogen and total phosphorus load estimations depend on flow and sediment conditions and are thus more event-based, while dissolved nutrient loads are more stable across the seasons. The sampling frequency recommended to give accurate load estimations for dissolved compounds (error < 5%) is weekly–biweekly sampling. A monthly sampling frequency produces a 20% error for yearly sediment load estimations using sediment-flow rating curves [52]. Other studies have shown that in order to produce average annual load estimates with less than 10% error, a 15 day sampling interval might be necessary for nitrate, 10 days

for soluble phosphorus and total phosphorus, and 5 days in the case of particulate phosphorus [53]. In light of this information, the sampling frequency of even the best group of catchments available in Hungary produced an error much higher than 10%. Other studies have suggested that up to 20% to 50% error can be expected [54], even more for suspended sediments [55]. Another problem is that the smaller the catchment size, the larger the estimation error might become.

The model performance seemed to be acceptable when compared to other studies [12,17,56]. Even though several low-accuracy monitoring stations were involved in the model validation, most of the stations were within a relatively uniform range of error along with the load scales. It seems that there was no clear relationship between the accuracy of the load measurements and the accuracy of the model estimations (Table 7). This might have been due to the fact that to improve the accuracy of measurements, an even higher number of measurements would be necessary, or that there were several sources of uncertainty in the model.

### 4.5. Local Emissions

Total emissions at small catchments had a recognizable spatial pattern for both nutrients (Figures 10 and 11). For total phosphorus, the pattern was easier to understand, as agricultural areas that are prone to erosion dominated the emissions. On the flatlands, however, where erosion is close to zero, phosphorus emission was also low. In the case of total nitrogen, emissions are strongly linked to nutrient surplus and groundwater emissions, while in larger cities, urban pathways also play an important role. Use of adjusted groundwater parameters resulted in a more even distribution of emissions between hilly and mountainous areas, while flatlands had smaller emissions generally due to smaller runoff and smaller surpluses.

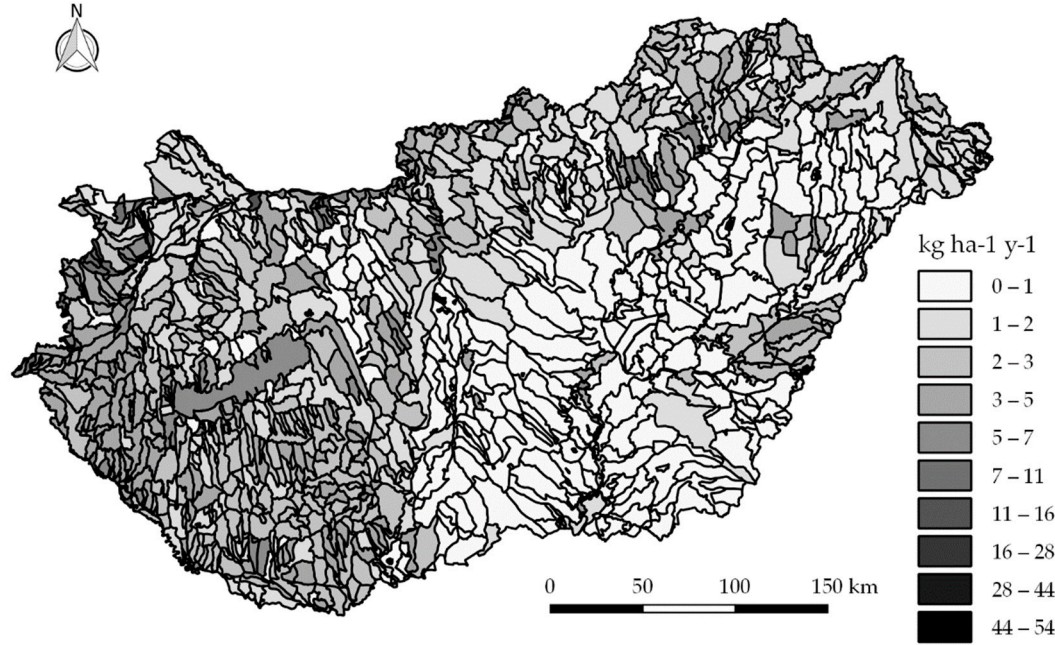

**Figure 10.** Diffuse-source area-specific TN emissions (totals of all pathways).

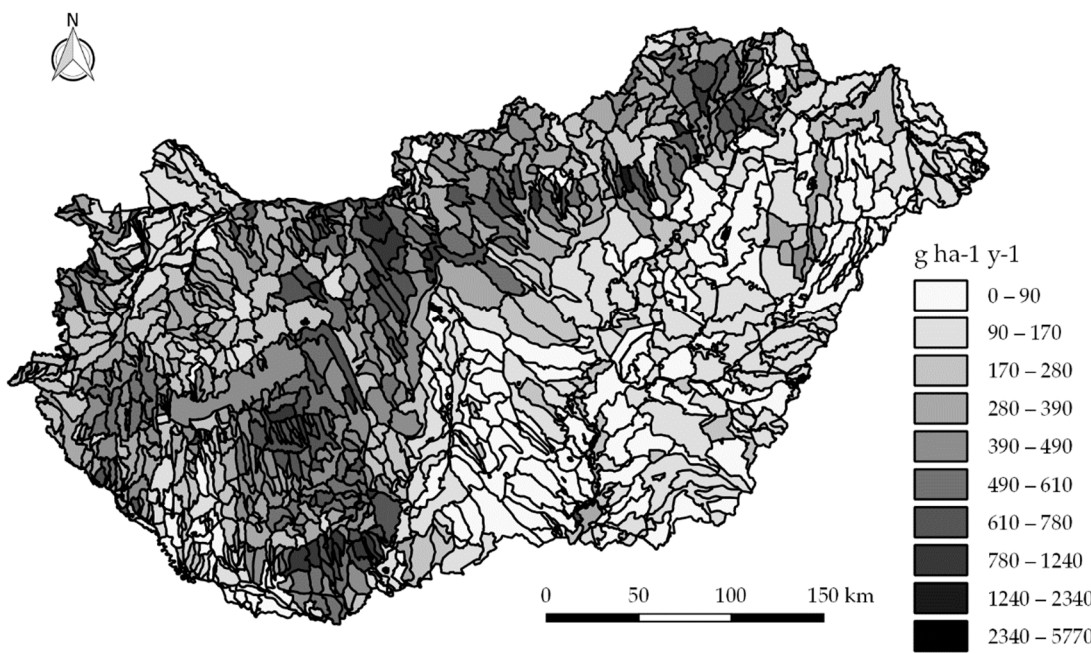

**Figure 11.** Diffuse-source area-specific TP emissions (sums of all pathways).

In terms of the distribution of nitrogen emissions between pathways (Figure 12), there was a very strong dominance of groundwater-related emissions (57%–70%) in all parts of the country. This agreed with previous studies from Germany [56] and the Danube River Basin [15]. Atmospheric deposition differs between slope classes due to the difference in surface water areas. Large lakes like Lake Balaton have huge emissions due to atmospheric depositions (660 t $y^{-1}$ from atmospheric deposition), as they are considered as part of the catchment. It should be noted, however, that the simple empirical relationship used in the current model might not be precise enough to describe the retention processes of large lakes. Agricultural erosion also differs among slope classes. This pathway was not considered significant in previous studies. However, in the current study, the higher values of organic matter and organic-matter-bound nitrogen caused higher values of erosion-bound losses in the medium slope category. The share of agricultural area decreased with higher slope, and agricultural erosion was thus less significant in the steepest slope category.

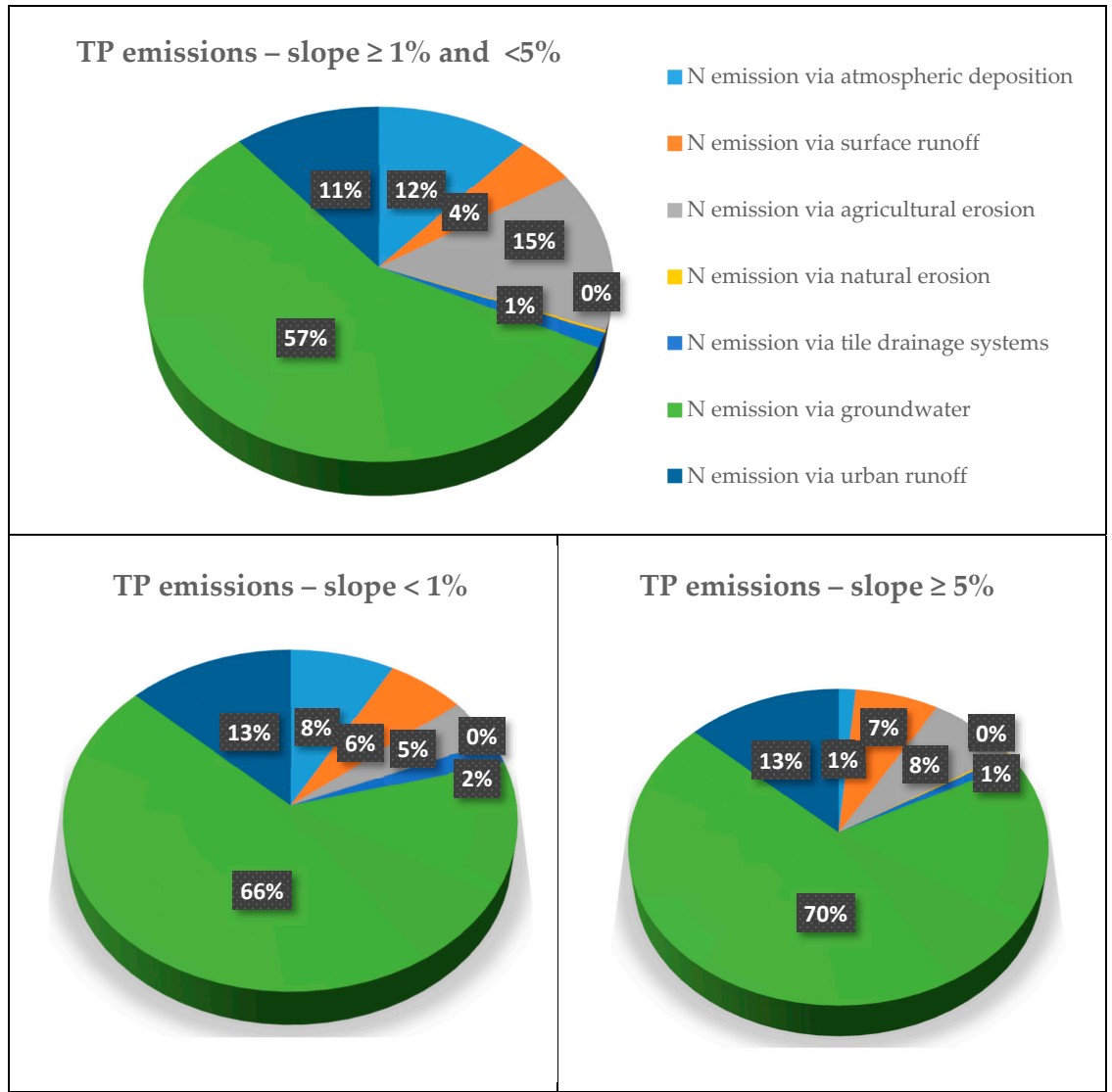

**Figure 12.** Division of total diffuse-source TN emission per pathway and slope class.

In agreement with previous model applications, the share of different pathways for total phosphorus differed largely from that for nitrogen. Agricultural erosion had a strong dominancy in all slope classes, but the magnitude of this dominancy differed between the slope classes (Figure 13). It is worth highlighting the significant portion of urban runoff in all slope classes. The third substantial pathway was groundwater, which was more critical in sandy catchments and wetlands than in clayey soils.

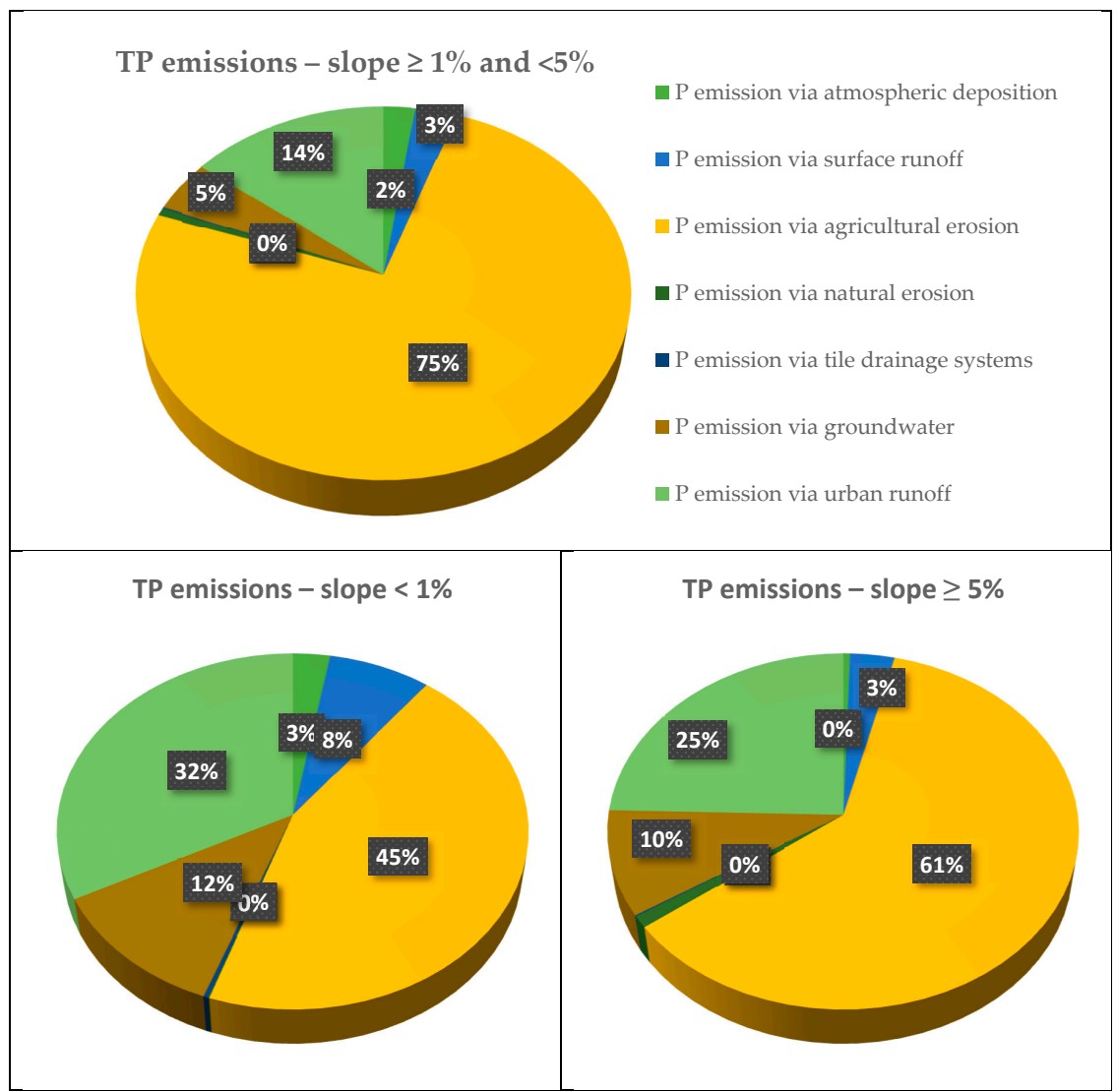

**Figure 13.** Division of total diffuse-source TP emissions per pathway and slope class.

*4.6. Comparing Nutrient Emission Results with Results from Previous Studies*

In this section, modeled emission values were compared to those of a previous model application. The previous application was delivered for the whole Danube Basin [15]. The average AU size in the previous study was 1660 km$^2$, while it was 86 km$^2$ in the present study.

A general remark regarding the large-scale application is that runoff patterns across Hungary seem not to follow the topography of the country. This can be explained by the relatively low specific runoff of this area compared to the Alps and the Carpathian Mountains, thus having a lower weight in the objective function during calibration. It was therefore proposed that further subregional flow monitoring data should be included in the calibration process of the next large-scale application of the model. In addition, the following differences between the results of both models should be mentioned.

Concerning nitrogen, surface runoff loads were 3.8 times higher than in the present study (Table 9). The larger-scale application calculated smaller erosion-bound emissions by much higher surface runoff. Without the precise knowledge of intermediate data for both of the models, it is difficult to see the exact reasons behind this anomaly. The large-scale application also estimated higher groundwater loads (Table 9). This was more than likely caused by differences in groundwater nitrogen retention parameters that caused higher concentrations in mountainous areas with higher specific runoff values.

**Table 9.** Total nitrogen and total phosphorus loads per pathway for the two modeled periods and the ICPDR application [15]. AD = Atmospheric deposition; SR = Surface runoff; AE = agricultural erosion; NE = natural erosion; TD = tile drainage; GW = groundwater; UR = urban runoff. ICPDR = the ICPDR application; HUN = the present study. Per. 1 = years 2009–2012; Per. 2 = years 2013–2016.

| | AD | SR | AE | NE | TD | GW | UR | Total Diffuse | Point Source | Total |
|---|---|---|---|---|---|---|---|---|---|---|
| **Total nitrogen loads (1000 t y$^{-1}$)** | | | | | | | | | | |
| ICPDR Per. 1 | 1605 | 3403 | 1006 | | 483 | 16,142 | 2527 | 25,168 | 7852 | 33 |
| HUN Per. 1 | 1646 | 894 | 3535 | 12 | 555 | 13,332 | 1876 | 21,849 | 10,314 | 32.2 |
| HUN Per. 2 | 1701 | 980 | 4597 | 18 | 314 | 11,053 | 2147 | 20,809 | 8629 | 29.5 |
| **Differences (%)** | | | | | | | | | | |
| HUN–ICPDR | 3 | −74 | *253* | | *15* | −17 | −26 | −13 | *31* | −2 |
| Per. 2–Per. 1 | 3 | 10 | *30* | 50 | −43 | −17 | *14* | −5 | −16 | −8 |
| **Total phosphorus loads (t y$^{-1}$)** | | | | | | | | | | |
| ICPDR Per. 1 | 0 | 19.1 | 773 | | 3.5 | 585 | 541 | 1922 | 1062 | 2986 |
| HUN Per. 1 | 60 | 127 | 1083 | 7 | 4 | 270 | 460 | 2010 | 1253 | 3264 |
| HUN Per. 2 | 60 | 123 | 1279 | 11 | 5 | 282 | 534 | 2294 | 1066 | 3360 |
| **Differences (%)** | | | | | | | | | | |
| HUN–ICPDR | | 565 | 41 | | 14 | −54 | −15 | 5 | 18 | 9 |
| Per. 2–Per. 1 | 0 | −3 | 18 | 57 | 25 | 4 | 16 | 14 | −15 | 3 |

Concerning phosphorus, surface runoff loads were only 15% of those in the present model (Table 9). Agricultural erosion was also lower, like in the case of nitrogen. Groundwater loads, in contrast, were 2.2 times higher.

Comparing the results of the two modeling periods, we concluded that changes in total emissions were not significant, but there were notable differences in the contribution of different pathways. Emission from point sources decreased by about 15% for both TN and TP. There was also a slight decrease in total diffuse nitrogen emission, while the total phosphorus emission was increased. This highlights the evidence of the nonlinear effects of the hydrological factors (e.g., a significant increase of erosion), which could not be clearly separated from the possible impact of the mitigation efforts (e.g., reducing soil nutrient balances.

## 5. Conclusions

The MONERIS model concept was applied to the surface water catchments of Hungary at the spatial scale set by the national river basin management plan. As it is important to have a model estimate on diffuse nutrient emissions that is as accurate as possible, some of the equations/parameters of the original model were reviewed and adjusted. These were

- the water balance equation with regard to the ratio of surface and subsurface runoff,
- the nitrogen retention parameters of the subsurface pathways (excluding the tile drainage),
- phosphorus concentrations in shallow groundwater,
- retention parameters for the retention in surface waters (rivers and lakes).

Even though some improvement in all the examined parts was achieved, not all of them proved to be significant.

Concerning water balance, the ratio of the surface and subsurface waters was improved when the digital-filter based separation was applied directly to catchments with monitoring points. After refinement of the water balance equation, load estimates were recalculated. It was found that the overall accuracy of model prediction did not change significantly, but the ratio between the pathways did change considerably. Surface runoff became more important with a larger share in the total emissions. This has consequences for mitigation actions, as the return period of subsurface waters was larger than that of surface pathways.

Subsurface nitrogen retention seemed to have a significant effect on model accuracy. In the current study, it was found that there were no big differences in subsurface nitrogen retentions between regions with higher recharge rates (northern and western mountains). Calibration of the retention parameters caused the drop of groundwater concentrations in the mountainous regions, while it slightly increased the concentrations in the lowlands of Hungary. Altogether, these changes improved the nitrogen load estimates across the whole calibration dataset, including catchments from all over the country.

Subsurface phosphorus concentrations were reviewed using all the available groundwater well data for shallow wells. Mean and median concentrations had large differences for clayey and sandy soils and forest, with a large portion of the values moving around the medians. For this reason, the concentration medians are proposed for use as representative values for soil or land use. The median values of Hungarian groundwater wells did not differ significantly from the original values except for sandy soils. Due to the higher infiltration rates and lower sorption capacity of sandy soils, the subsurface phosphorus concentrations were more sensitive to the surplus history of the region under study. Therefore it is recommended that groundwater well data be checked regionally.

Surface water nutrient retention turned out to be the most important part of the model. While in the current application, the nitrogen retention parameters did not improve the model accuracy significantly, adjustment of phosphorus retention parameters improved the overall model performance by 20%–30%.

The comparison of the present calculations with the larger-scale application of the same model led to the conclusion that the accuracy of total load estimates differed in the distribution of the loads among pathways. The difference of total diffuse loads was primarily caused by the insufficient spatial representation of runoff in the larger scale application. Therefore it is recommended that in the future, a more rigorously reviewed network of monitoring stations be used for flow calibration.

**Author Contributions:** Conceptualization, Z.J.; writing—original draft preparation, Z.J.; writing—review and editing, M.K.K. & A.C; visualization, M.K.K. & Z.J.; supervision, A.C.; funding acquisition, A.C. All authors have read and agreed to the published version of the manuscript.

**Funding:** The research reported in this paper was supported by the Higher Education Excellence Program of the Ministry of Human Capacities in the frame of the Water sciences & Disaster Prevention research area of BME (BME FIKP-VÍZ). Financial support from the General Directorate of Water Management Hungary is greatly acknowledged.

**Acknowledgments:** The authors thank Márta Bagi, István Bíró, Szabina Pelyhe, and György István Tóth for the preparation of the monitoring databases of wastewater, surface, and subsurface water quality.

**Conflicts of Interest:** The authors declare no conflict of interest.

## Appendix A  Input Data Tables

**Table A1.** Spatial data used for the MONERIS model input.

| Data Class | Detail | Data Source and Method | Source/Comment |
|---|---|---|---|
| Catchment area | | GIS data for RBMP 2015 | [27] |
| Land use data | Urban area, arable land areas in slope classes, grassland, woodland, shrubland, water surface area, mines, open areas, wetlands | Spatial statistics based on GIS analysis of CORINE LAND COVER grid | [28] |
| Soil classification | Sandy agricultural soils<br>Loamy agricultural soils<br>Silty agricultural soils<br>Clayey agricultural soils | The classification made based on national soil texture database | [33] |
| | Fen agricultural soils<br>Bog agricultural soils | Agrotopo fen type<br>Agrotopo bog type | [27] |
| Underlying geology | Unconsolidated rock areas near groundwater<br>Unconsolidated rock areas far groundwater | Statistics were based on raster data, combining Agrotopo database and groundwater table depth from RBMP 1 in 2009 | [29] |
| | Solid rock areas with good porosity<br>Solid rock areas with poor porosity | Agrotopo based statistics<br>Agrotopo based statistics | [29] |
| Average elevation | | Based on a 50 m resolution hydrodem raster | [26] |
| Average slope | | Based on a 50 m resolution hydrodem raster | [26] |

**Table A2.** Temporal data used for the MONERIS model.

| Data Class | Detail | Input Data and Method | Comment |
|---|---|---|---|
| Net total runoff | Average riverbed runoff in the modeled period | GDWM long term average runoff data | Corrected by rainfall ratios and measured discharge values, where available |
| Average temperature | Average water temperature for the modeled period | FEVISZ database [57] | |
| Yearly precipitation current | Average yearly precipitation in the model period | Precipitation monitoring network data operated by GDWM | Precipitation distribution by Thiessen polygon method. |
| Summer half-yearly precipitation current | Average summer half-year precipitation in the modeled period | Precipitation monitoring network data operated by GDWM | |
| Summer half-yearly precipitation long term | Average summer half-year precipitation in between 1981–2010 | Precipitation monitoring network data operated by GDWM | |
| Evapotranspiration | Mean annual evapotranspiration of the catchments | Country scale evapotranspiration distribution map 2001–2009 | Data source: [22] |
| Measured yearly TN load | Product of yearly average water quality data and yearly average of measured discharge data | FEVI database [57] | |
| Measured yearly DIN load | | | |
| Measured yearly TP load | | | |
| NH4-N, NO3-N deposition rate-current | Yearly average deposition rates of nutrients in different forms. (mg m$^{-2}$ y$^{-1}$) | EMEP European air quality data maps for each component | [38] |
| NH4-N, NO3-N deposition rate-long term | | | |
| TP deposition rate | Default value by MONERIS | | |
| Soil loss from agricultural areas, grasslands and natural covered land | Average annual soil loss per land use (t ha$^{-1}$ y$^{-1}$) | JRC maps & USLE method [30] | K factor [58]<br>K factor (2013–2016) [32]<br>R Factor [59] |
| N-content of topsoil | Total N content of the topsoil (mg kg$^{-1}$) | AGROTOPO [29] & DOSOREMI [60] databases | Calculated based on soil organic carbon content |
| N-surplus-current | Nitrogen surplus in the topsoil for the 2009–2012 and 2013–2016 periods (kg ha$^{-1}$ y$^{-1}$) | County scale statistics of N balance (average of 2009–2012 and 2013–2016 periods) [36] | |

**Table A2.** *Cont.*

| Data Class | Detail | Input Data and Method | Comment |
|---|---|---|---|
| N-surplus-residence time | Nitrogen surplus in the topsoil for the average groundwater residence time of the catchments (kg ha$^{-1}$ y$^{-1}$) | County scale statistics of N balance for the 1961 to 2016 period [36] | |
| Accumulated P-surplus | Phosphorus surplus in the topsoil from 1961 to 2016 (kg ha$^{-1}$ y$^{-1}$) | Large regional, later county scale statistics of P balance for the entire country | Catchment average is calculated based on the country scale raster |
| External discharge | Yearly average river Q data at the upper boundary of a river transboundary sub-catchment Period: 2009–2012 and 2013–2016 | Data derived from the discharge data of the national river monitoring network | Some large rivers lack satisfactory Q and/or WQ data (Vág, Garam, Ipoly tributaries etc.) TNMN is used [61] |
| External TN load | Yearly average TN load at the upper boundary of a transboundary sub-catchment | FEVI database [57] | Some large rivers lack satisfactory Q and/or WQ data (Vág, Garam, Ipoly tributaries etc.), TNMN is used [61] |
| External DON load | Yearly average DON load at the upper boundary of a transboundary sub-catchment | FEVI database [57] | Some large rivers lack satisfactory Q and/or WQ data (Vág, Garam, Ipoly tributaries etc.) [61] |
| External TP load | Yearly average TN load at the upper boundary of a transboundary sub-catchment | FEVI database [57] | Some large rivers lack satisfactory Q and/or WQ data (Vág, Garam, Ipoly tributaries etc.) [61] |

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
