# Peer review of "Modification of the MONERIS Nutrient Emission Model for a Lowland Country (Hungary) to Support River Basin Management Planning in the Danube River Basin"

_water, doi:10.3390/w12030859_

Round 1
Reviewer 1 Report
1111/5000 Paper has a very interesting topic.The problem of nutritionists is very topical, but due to the lack of clarity of resources, their value is difficult.
The text describes the procedure for evaluating the territory of Hungary.
The text does not describe what DIN is - probably dissolved inorganic nitrogen.
I have some questions about the text to improve the evaluation of results.
Why there are very different results for Zagyva-cr. (upper) page 11?
Table 5. Model constants before and after adjustment have very significant differences. For example, the value of Solid Rock, Poor Porosity was adjusted from 78.54 to 99787. And then, UC Rock, Deep GW from 68560 to 7917. Is such a fundamental adjustment of values ​​to the signs of model insensitivity?
What is the confidence in the results of such fundamental model modifications?
Total nitrogen and total phosphorus loads also have significant surface and groundwater treatments.
Subsequently, the results are shown in Table 9. The deviations from the original model are 253 or 565%.
It would be good to add the text to make the results clearer. Maybe replace the lengthy tables in the Annex. 1111/5000 Paper has a very interesting topic.
The problem of nutritionists is very topical, but due to the lack of clarity of resources, their value is difficult.
The text describes the procedure for evaluating the territory of Hungary.
The text does not describe what DIN is - probably dissolved inorganic nitrogen.
I have some questions about the text to improve the evaluation of results.
Why there are very different results for Zagyva-cr. (upper) page 11?
Table 5. Model constants before and after adjustment have very significant differences. For example, the value of Solid Rock, Poor Porosity was adjusted from 78.54 to 99787. And then, UC Rock, Deep GW from 68560 to 7917. Is such a fundamental adjustment of values ​​to the signs of model insensitivity?
What is the confidence in the results of such fundamental model modifications?
Total nitrogen and total phosphorus loads also have significant surface and groundwater treatments.
Subsequently, the results are shown in Table 9. The deviations from the original model are 253 or 565%.
It would be good to add the text to make the results clearer. Maybe replace the lengthy tables in the Annex.
Reviewer 2 Report
Jolánka et al.s manuscript titled "Adjustment of the MONERIS nutrient emission model for a lowland country to support River Basin Management Planning" is a well-done scientific study that applies a well-known watershed nutrient model to the Hungarian portion of the Danube basin. The manuscript is clear as to all the model parameters, measures of fit etc. Further, it is well written and should be of interest to readers of Water. There are a few minor comments/ corrections below that will improve the manuscript. After the necessary changes are made I recommend this work for publication.
Minor Comments
Title- Perhaps change "Adjustment" to "Modification" and add "(Hungary)" after "lowland country" and add "in the Danube River watershed" to the end. Thus the new title would be- "Modification of the MONERIS nutrient emission model for a lowland country (Hungary) to support River Basin Management Planning in the Danube River watershed"
Line 52- Replace "handy" with "useful"
Line 106- Replace "the Balaton" with "Lake Balaton"
Line 186- Replace "wider" with "longer"
Line 347- Give p-value for regression line mentioned here.
Figure 6- Tell readers what a "violin plot" shows them.
Line 535- Remove "the" before those
Reviewer 3 Report
1) The work tried to revise some parameters from the MONERIS nutrient emission model toward three objectives including degree of MONERIS capacity estimating nutrient fluxes, identification of the model components and assessment of nitrogen and phosphorus for application on water contamination in the Hungary region of Danube River Basin. However, we can not find the clear idea of the works in the manuscript. The authors should indicate clearly which parameters/components were improved in the MONERIS and the reasons of the modification, model simulation and validation, and suggestion for river basin management planning. In addition, it is important how the research support river basin management planning, nevertheless, it seems that the part is missing in the manuscript.
2) The structure of the manuscript needs a big adjustment, especially, model description, validation. In addition, which components of the model are revised, which needs the clearer description. Some hydrological variables should be expressed clearly by means of further understanding the physical processes on hydrological cycle and pollutants diffusion and movement, for example, runoff from precipitation, surface runoff, runoff from snowmelt, etc. And, it seems that equation (1) and (2) has a conflict in the section 2.2.1. Moreover, The content on model validation appears in the result and discussion.
3) It is difficult that some statements are understood clearly, accordingly, the language needs a higher quality polishing.
Special comments:
1) Line 91: How does the degree to which the model system MONERIS is capable… be quantified?
2) Line 142: Is there a conflict between Equation (1) and (2)?
3) Line 147: In the equation (2), coefficient kw2 should be kw2, ‘w2’ should be subscript.
4) Line 148: The parameters that appear in the equations need the unit description, and what does some constant values, such as kw1,kw2,…
5) Line 216: Was the universal soil loss equation integrated into the MONERIS model?
6) Line 221: Was the OECD method also integrated in the MONERIS model? And, it need a full name as OECD first appears.
7) Line 224: Fig.2 (a) requires a legend for better understanding the implication of each curve.
8) Line 231: It needs a full name of EMEP that first appears.
9) Line 284: Order number of the cited equation is incorrect, not Eq. 3.
10) Line 310: The name of the coefficients including nutrient retention parameters (8) and (9) are better than citation of them.
11) Line 315: The model validation is not clear, which is needed to prove the reliability of the improved MONERIS for Hungary region.
12) Line 344: The part should be moved to section 2.2 on the model description.
13) Line 360: In the sentence “it was selected….”, does the model refer to the MONERIS model?
14) Line 369: To understand the Fig. 6 is difficult, which needs the more detailed description.
15) Line 400: Model validation should be moved to the section 2.5.4.
16) Line 414: What is the full name of BFI?
17) Line 443: What implication of the analysis? What is the role of the modification of the MONERIS model?
Round 2
Reviewer 3 Report
Most questions have been replied and corresponding content has been revised in new manuscript. Nevertheless, there are three minor suggestions.
1) For the model validation parts in discussion, if author thought that these contents are important points for description of modification of MONERIS model, they maybe stay location, yet, it is better for readers' understanding to rename the subtitle.
2) In the model description, are the equations (1-4) original express of the MONERIS? In addition, model input items are not just Q and Qww, right?
3) For the question 1), "determine the degree to which the model system MONERIS is capable of estimating nutrient fluxes in a lowland country such as Hungary" is one of your study objectives.
Author Response
Thank you again for your remarks, we think that these are relevant, therefore we have made some modifications as listed below:
1) For the model validation parts in discussion, if author thought that these contents are important points for description of modification of MONERIS model, they maybe stay location, yet, it is better for readers' understanding to rename the subtitle.
The subtitle has been renamed to "Model validation in the context of the literature" in line 492
2) In the model description, are the equations (1-4) original express of the MONERIS? In addition, model input items are not just Q and Qww, right?
Yes, this section gives a description of the original model equations. Yes, Q and Qww are model inputs, the rest of the water balance components are estimated by the model as described by eq 2-4. Q atm is calculated from precipitation data and water surface. There is additional input data (land use for example) that is necessary for these calculations, but we thought there is no need to give a full description as they are not that relevant to our methodology.
3) For the question 1), "determine the degree to which the model system MONERIS is capable of estimating nutrient fluxes in a lowland country such as Hungary" is one of your study objectives.
Yes, thank you, this is why we dealt with the validation in high detail with datasets from two subsequent periods and this is why we included it in the discussion as well.
